# Surface Functionalization of Magnetic Nanoparticles Using a Thiol-Based Grafting-Through Approach

**Philip Biehl and Felix H. Schacher *** 

Laboratory for Organic and Macromolecular Chemistry, Institut für Organische Chemie und Makro-molekulare Chemie, Friedrich-Schiller-Universität, Jena IOMC Humboldtstr. 10, D-07743 Jena, Germany; philip.biehl@uni-jena.de

* Correspondence: felix.schacher@uni-jena.de

**Abstract:** Here we describe a simple and straightforward synthesis of different multifunctional magnetic nanoparticles by using surface bound thiol-groups as transfer agents in a free radical polymerization process. The modification includes a first step of surface silanization with (3-mercaptopropyl)trimethoxysilane to obtain thiol-modified nanoparticles, which are further used as a platform for modification with a broad variety of polymers. The silanization was optimized in terms of shell thickness and particle size distribution, and the obtained materials were investigated by dynamic light scattering (DLS), thermogravimetric analysis (TGA), transmission electron microscopy (TEM), and energy-dispersive X-ray spectroscopy (EDX). Subsequently, the free radical polymerization of different monomers (*tert*-butyl acrylate (*t*BA), methyl methacrylate (MMA), styrene, 2-vinyl pyridine (2VP), and *N*-isopropylacrylamide (NIPAAm)) was examined in the presence of the thiol-modified nanoparticles. During the process, a covalently anchored polymeric shell was formed and the resulting core–shell hybrid materials were analyzed in terms of size (DLS, TEM), shell thickness (TGA, TEM), and the presence of functional groups (attenuated total reflectance Fourier-transform infrared spectroscopy (ATR-FT-IR)). Hereby, the shell leads to a different solution behavior of the particles and in some cases an increased stability towards acids. Moreover, we examined the influence of the nanoparticle concentration during polymerization and we found a significant influence on dispersity of the resulting polymers. Finally, we compared the characteristics of the surface bound polymer and polymer formed in solution for the case of polystyrene. The herein presented approach provides straightforward access to a wide range of core–shell nanocomposites.

**Keywords:** magnetic multicore nanoparticles; surface modification; organic–inorganic nanocomposites; grafting-through

## 1. Introduction

Magnetic nanoparticles (MNPs) are discussed for a rather broad variety of applications such as magnetic resonance imaging (MRI) [1], as drug delivery systems [2,3], for tumor hyperthermia [4], bio separation [5–7], water purification systems [8–10], and as catalytic supports [11,12]. All these fields require well-controlled surface functionalization and a defined MNP morphology to ensure certain physical characteristics like magnetic properties or nanoparticle size while at the same presenting the desired chemical functionality at the surface. Especially the latter depending on the environment in which the particles will be applied and, here, the range includes highly saline aqueous media [7] as well as organic solvents [13]. Therefore, surface modification of MNPs can be regarded as one of the key elements for all applications being discussed. Here, the introduction of polymer surface coatings has been proven to impart enhanced suspension stability [14], protein repellence [15], solubility in a diverse set of environments [13,16], and adjustable surface charge [17]. For instance, both shielding

and anti-fouling can be ascribed to poly(ethylene glycol)-based (PEG) surfaces as they are able to bind large amounts of water [18]. Control over any type of interactions with other molecules are another aspect for choosing a suitable surface coating as for example in biomedical applications a binding of biomolecules can be undesirable [19], while during bio sorting selective binding of certain partners is crucial [20].

There are several strategies for the immobilization of polymers at the surface of MNPs which can roughly be divided into adsorptive or covalent binding [21]. Whereby covalent binding itself can be subdivided into grafting-to, grafting-from, and grafting-through approaches, in any case the surface needs to be equipped with suitable binding motifs. In this regard, silane coupling agents are one of the most prominent ways to install certain functional groups, ranging from thiols over amines to vinyl moieties [7,22–26]. Among those, thiols are of interest for subsequent protein binding, with regard to thiol-ene chemistry but also in the context of radical polymerization techniques. They are widely employed as chain transfer agents in reversible addition fragmentation chain transfer polymerization RAFT [27] or the Strathclyde method [28,29] and typically exhibit rather high transfer rates. Such strategies have already been employed for the formation of polymer coatings at surfaces [30–35]. Here, mostly silica is used as starting material, either as macroscopic glass slides or as nanoparticles [30,32,34,35], and only few studies investigate MNP as starting material [31,33]. Whereas the general concept of immobilizing polymeric surface coatings using thiol moieties as anchoring points is known, our aim was to use this as a general strategy for the preparation of core–shell nanomaterials with a broad variety of surface functionality. Hence, we present a straightforward method to access MNP with different polymeric shells (poly(*tert*-butyl acrylate) (P*t*BA), poly(methyl methacrylate) (PMMA), polystyrene (PS), poly(2-viylpyridine) (P2VP), and poly(*N*-isopropyl acrylamide) (PNIPAAm)) starting from thiol-functionalized MNP as reactive platform. In a first step, pristine MNP are modified using silanization to immobilize thiol-groups and these particles are subsequently used as chain transfer agents in free radical polymerization (FRP) of different monomers. As a side effect, some polymers will be covalently linked to the particle surface, representing a simple one-step surface modification where the obtained nanocomposites can be magnetically separated from the reaction solution. Besides surface anchorage, this strategy also allows for a rapid polymer characterization in terms of dispersity and molecular weight as we assume that both polymers formed in solution and immobilized at the surface feature comparable molecular characteristics.

## 2. Materials and Methods

### 2.1. Instruments

Dynamic light scattering: DLS measurements were performed using an ALV Laser CGS 3 goniometer equipped with a 633 nm HeNe Laser (Langen, Germany) at 25 °C and at a detection angle of 90°. The CONTIN algorithm was used to evaluate the obtained data.

Thermogravimetric analysis: The samples (prepared and washed as described in the Nanoparticle Coating section) were magnetically separated and freeze-dried for 72 h. TGA measurements were carried out from 30 to 800 °C under synthetic air within a heating range of 10 K/min in a PerkinElmer TGA8000 device (Waltham, MA, USA).

Transmission electron microscopy: For TEM from aqueous solutions, copper grids were rendered hydrophilic by argon plasma cleaning for 120 s (Diener Electronics, Ebhausen, Germany). A total of 15 µL of the respective sample solution was applied to the grid, and excess sample was blotted away with a filter paper. TEM images were acquired with a 200 kV FEI Tecnai G 2 20 microscope equipped with a 4 K × 4 K Eagle HS CCD and a 1 K × 1 K Olympus MegaView camera for overview images (Hillsboro, OR, USA).

Size exclusion chromatography: SEC measurements were performed on an Agilent system (Santa Clara, CA, USA) equipped with G1310A pump, a G1362A refractive index detector, and both a PSS Gram30 and a PSS Gram1000 column in series (Mainz, Germany). DMAc + 0.21 wt % LiCl was applied

as eluent at 1 mL min$^{-1}$ flow rate and the column oven was set to 40 °C. For the calibration a PMMA or PS standard was used.

ATR-IR-spectroscopy: Measurements were performed on a PerkinElmer Frontier FT-IR, NIR, and FIR Spectrometer equipped with Golden Gate Single Reflection spectrometer (Waltham, MA, USA).

Ultrasonication: Ultrasonication was performed using an ElmaSonic S30H ultrasonic unit (Singen, Germany) and by using an ultrasound processor: Sonics VibraCell VC505 (Newtown, CT, USA).

Syringe pump: For defined addition of liquids, a Landgraf Laborsysteme HLL LA-30 syringe pump (Langenhagen, Germany) was used.

## 2.2. Materials

Sodiumhydrogencarbonate, iron(II) chloride, iron(III) chloride, sodium hydroxide, and potassium hydroxide were obtained from Roth in p.a. grade and used without further purification. MPTS, guanidine hydrochloride, 2-vinylpyridine, methyl methacrylate, *tert*-butyl acrylate, and styrene were purchased from Merck and all monomers were purified by column chromatography (AlOx) to detach stabilizer prior to use. 1 M solution of hydrogen chloride in ethyl acetate, absolute ethanol were purchased from Thermo Fisher Scientific and used as received. Tetrahydrofuran was of technical grade and distillated prior to use. The photoinitiator Lucirin-TPO was kindly provided by BASF.

## 2.3. Synthesis

Synthesis of MCNP: Multicore iron oxide nanoparticles were prepared according to previous work by Dutz et al. [36]. Briefly, the particles were synthesized by slowly adding a 1 M NaHCO$_3$ solution to a FeCl$_2$/FeCl$_3$ solution (total iron concentration 1.25 M; Fe$^{2+}$/Fe$^{3+}$ ratio = 1/1.3) at a rate of 0.9 mL/min under permanent stirring up to pH 8, leading to the formation of a brownish precipitate. Afterward, the solution was boiled for 5 min to form an almost black precipitate. The MCNP were then magnetically washed with distilled water until a conductivity below 10 μS/cm (room temperature) was reached.

Synthesis of MPTS@MCNP: In a typical procedure MCNP (100 mg) were suspended in 500 mL water (0.2 mg mL$^{-1}$) with a pH of 11 (adjusted by NaOH) in a two neck round bottom flask. An ultra-sonication finger (1 min with 30% intensity) was used to suspend the particles and the suspension was directly stirred mechanically with 250 rpm by a Teflon stirrer and treated with an ultra-sonication bath. An amount of MPTS (100, 200, 300, or 400 μL) was dissolved in dry ethanol with a final concentration of 90 mmol/L. The solution was added with a constant drop rate of 50 μL/min to the nanoparticle suspension. Subsequently, the suspension was stirred for 19 h and the particles were afterwards separated magnetically followed by four washing steps with 40 mL water.

Synthesis of Polymer@MPTS@MCNP: In a typical procedure MPTS@MCNP (10 mg) were separated from solution and treated with 1 M guanidine hydrochloride solution (1 mL) for one hour. Afterwards, the particles were washed three times with THF. A microwave vial was charged with the respective monomer (48 mmol), TPO (13.5 mg), and 20 mL THF, sealed and degassed for 20 min before the suspension was treated with ultrasonication. The reaction vial was exposed to intense UV-irradiation for 15 min without stirring, to avoid unwanted magnetic precipitation. The obtained particles were separated magnetically and the supernatant was kept for further analysis. The particles were washed three times with THF to remove any unbound polymer and dried under vacuum.

Detachment of PS from Particles: 5 mg PS@MPTS@MCNP were suspended in a 1 M solution of hydrogen chloride in ethyl acetate. After several minutes the dissolving of particles was observed and a yellow solution (Fe$^{3+}$) was obtained. The yellow solution was separated from a resulted precipitate by centrifugation. A 1 M aqueous solution of KOH was added to the precipitate and sonicated for 30 min. After separation of the solution from the precipitate the obtained PS was dissolved in DMAc and measured by SEC.

## 3. Results and Discussion

### 3.1. MPTS@MCNP

In this study, we used multicore magnetic nanoparticles (MCNP) which were synthesized by coprecipitation of a $Fe^{2+}/Fe^{3+}$-solution as described earlier by Dutz et al. [36,37]. The particles exhibited a radius of 26 nm and consisted of several primary cores of about 11 nm which build up a cluster-like structure (multicore particle). This structure was already present before any coating process (TEM micrographs Figure S1, Supplementary Materials) and the average particle size was just below the single domain radius of iron oxides (Dc ($Fe_3O_4$) = 128 nm) [38]. Both size and magnetic properties of the MCNP used here would be favorable for an application in the field of hyperthermia or any strategy aiming for heat generation or rapid magnetic separation [39]. As MCNP tend to undergo secondary aggregation due to strong magnetic dipolar–dipolar interactions [40], the first challenge was the introduction of a defined silane coating while maintaining a well-dispersed system. The formation of a silica shell around MNPs is well-studied and already shown for many different systems [22,25,41–43]. However, we found that several protocols are difficult to adopt if the nanoparticles change in size or from single to multicore character or in chemical composition which made it necessary to optimize the coating procedure for the herein used particles. Several protocols suggest a two-step route for surface functionalization using 3-(mercaptopropyl)trimethoxysilane (MPTS, Scheme 1) [24,44,45] by applying first tetraethyl orthosilicate, followed by the functional silane. Nevertheless, in our case a straightforward one-step procedure turned out to lead to well-defined core–shell nanoparticles. Therefore, a diluted suspension of MCNP (0.2 mg/mL) at a pH of 11 (adjusted with KOH) was stirred mechanically and MPTS in dry ethanol was added at a constant flow rate of 50 μL min$^{-1}$. Higher concentrations of nanoparticles led to aggregation and thus to enclosure of bigger agglomerates within the same siloxane shell. Dilution was the only way we found to avoid larger aggregates during the coating process. After addition of the silane, the mixture was allowed to react for 18 h. The particles were then magnetically separated and washed three times with water. After application of the siloxane shell, the suspension changed color from dark to lighter brown, a phenomenon which is also described for similar coating procedures in the literature [41].

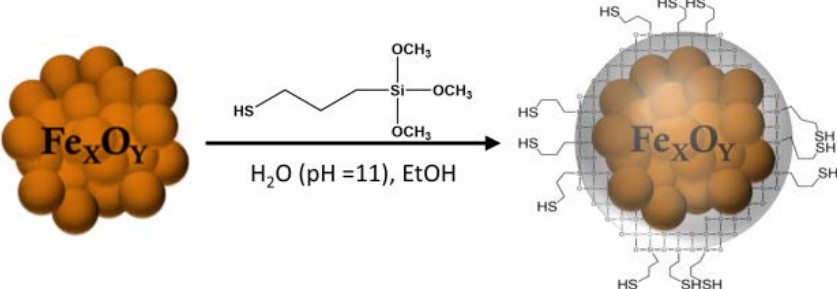

**Scheme 1.** Surface functionalization of magnetic multicore nanoparticles (MCNP) with (3-mercaptopropyl) trimethoxysilane (MPTS).

The ratio of MCNP to MPTS was optimized in order to control both size and shell thickness of the siloxane shell. By varying the amount of MPTS we were able to obtain different shell thicknesses and a rather good control about the overall size of the resulting nanoparticles. Figure 1A shows the size distribution measured by dynamic light scattering (DLS) prior to and after the coating process in water. The formation of a siloxane shell led to an increased hydrodynamic radius and clearly broadened the size distribution. While the pristine particles had a hydrodynamic radius of 26 nm, the core–shell particles exhibited an increasing hydrodynamic radius from 50 up to 100 nm as the ratio of MPTS to MCNP was increased from 1:1 to 4:1. Furthermore, we used thermogravimetric analysis (TGA) to quantify the amount of MPTS at the nanoparticle surface (Figure 1B) due to the presence

of organic compounds (–(CH$_2$)$_3$–SH groups). The pristine MCNP showed a weight (wt) loss of 2.6% which is attributed to remaining carbonates which were used during the MCNP synthesis. The organic compounds in MPTS@MCNP led to an additional weight loss between 200 °C and 700 °C. With an increasing amount of MPTS the weight loss increased from 8.2% (ratio 1:1) up to 27.8% (ratio 4:1). This is a strong indication for the presence of a siloxane shell at the nanoparticle surface.

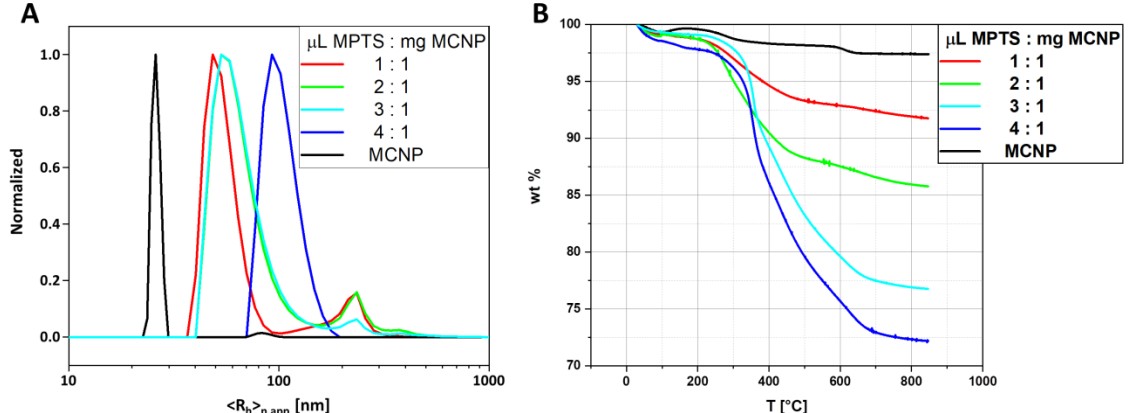

**Figure 1.** (**A**) Number-weighted DLS CONTIN plots of pristine MCNP (black line, ⟨R$_h$⟩$_{n,app}$ = 26 nm), and MPTS@MCNP obtained for varying ratios of MPTS to MCNP of 1:1 (red line, ⟨R$_h$⟩$_{n,app}$ = 48 nm), 2:1 (green line, ⟨R$_h$⟩$_{n,app}$ = 53 nm), 3:1 (cyan line, ⟨R$_h$⟩$_{n,app}$ = 52 nm), 4:1 (blue line, ⟨R$_h$⟩$_{n,app}$ = 93 nm). (**B**) Thermograms of MCNP (black line, 2.6% overall weight loss), MPTS@MCNP obtained from a ratio of MPTS to MCNP of 1:1 (red line, 8.2% overall weight loss), 2:1 (green line, 14.2% overall weight loss), 3:1 (cyan line, 23.2% overall weight loss), 4:1 (blue line, 27.8% overall weight loss).

Transmission electron microscopy (TEM) was used to confirm the mean size of the MPTS@MCNP nanoparticles (Figure 2), resulting in approximately 100–200 nm diameter which is in good agreement with the results from DLS measurements. With an increasing amount of MPTS a light grey shell becomes visible around the MCNP core, mainly evident for MPTS:MCNP ratios of 3:1 and 4:1. It was further possible to adjust different shell thicknesses, e.g., around 5 nm for the 3:1 ratio while about 15 nm were found in case of 4:1. As according to this data the MPTS@MCNP using a 3:1 ratio exhibited a defined and reasonably thin shell, these samples were chosen for further modifications.

In addition, scanning transmission electron microscopy/energy-dispersive X-ray spectroscopy (STEM-EDX) mapping was performed to verify the presence of thiols. The sample was analyzed towards iron-, silica-, sulfur-, and oxygen-content and—as can be seen from Figure 3—all elements are relatively homogeneously distributed. Here, we analyzed a rather large particle agglomerate as the performance of STEM-EDX measurements is improved—however, the data confirms successful immobilization of thiol-groups at the surface of MPTS@MCNP.

### *3.2. Polymer@MPTS@MCNP*

The thiols at the surface of MPTS@MCNP enable further functionalization as they can be used as chain transfer agents in radical polymerization processes [28,46,47]. As side reaction, some polymer chains will be covalently linked to the MPTS@MCNP surface. Prior to use, MPTS@MCNP were treated with an aqueous solution of 1 M guanidine hydrochloride to activate the thiols followed by three washing steps with THF (the solvent for all polymerizations). THF was chosen as it is a suitable solvent for all investigated monomers/polymers and furthermore allowed a well dispersed nanoparticle suspension without aggregation occurring (Figure S2A, Supplementary Materials). It was possible to polymerize a broad variety of vinyl monomers (*t*BA, MMA, styrene, 2VP, NIPAAm) in the presence of the particles (Scheme 2). We chose polymer coatings with different properties in order to show the broad applicability of our strategy. All polymerizations were carried out under comparable conditions concerning monomer concentration and monomer to initiator ratio. Thus, each attempt consisted

of 10 mg nanoparticles in 20 mL THF, diphenyl(2,4,6-trimethylbenzoyl)phosphine oxide (TPO) as photo-initiator with a concentration of 1.9 mmol $L^{-1}$ and the respective monomer at a concentration of 2.4 mol $L^{-1}$ (Table 1).

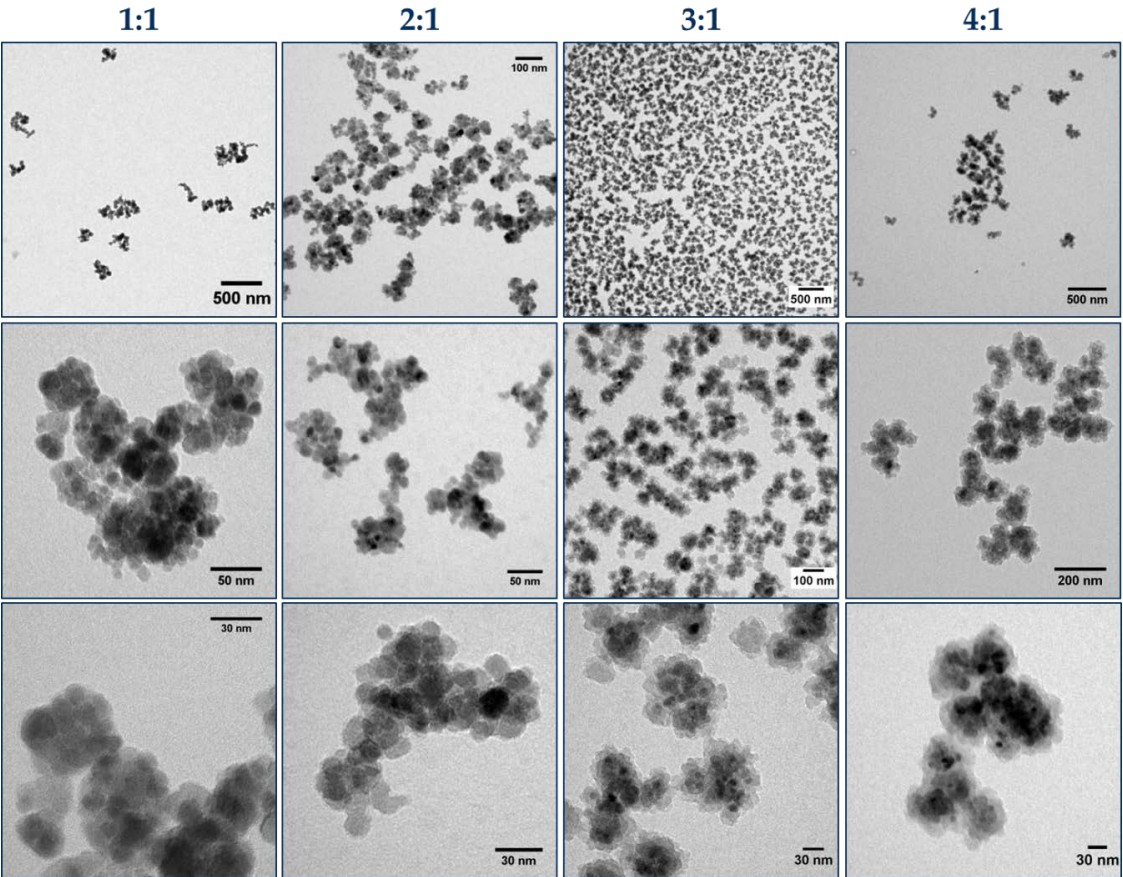

**Figure 2.** TEM micrographs of MPTS@MCNP obtained from different ratios of MPTS to MCNP at different magnifications.

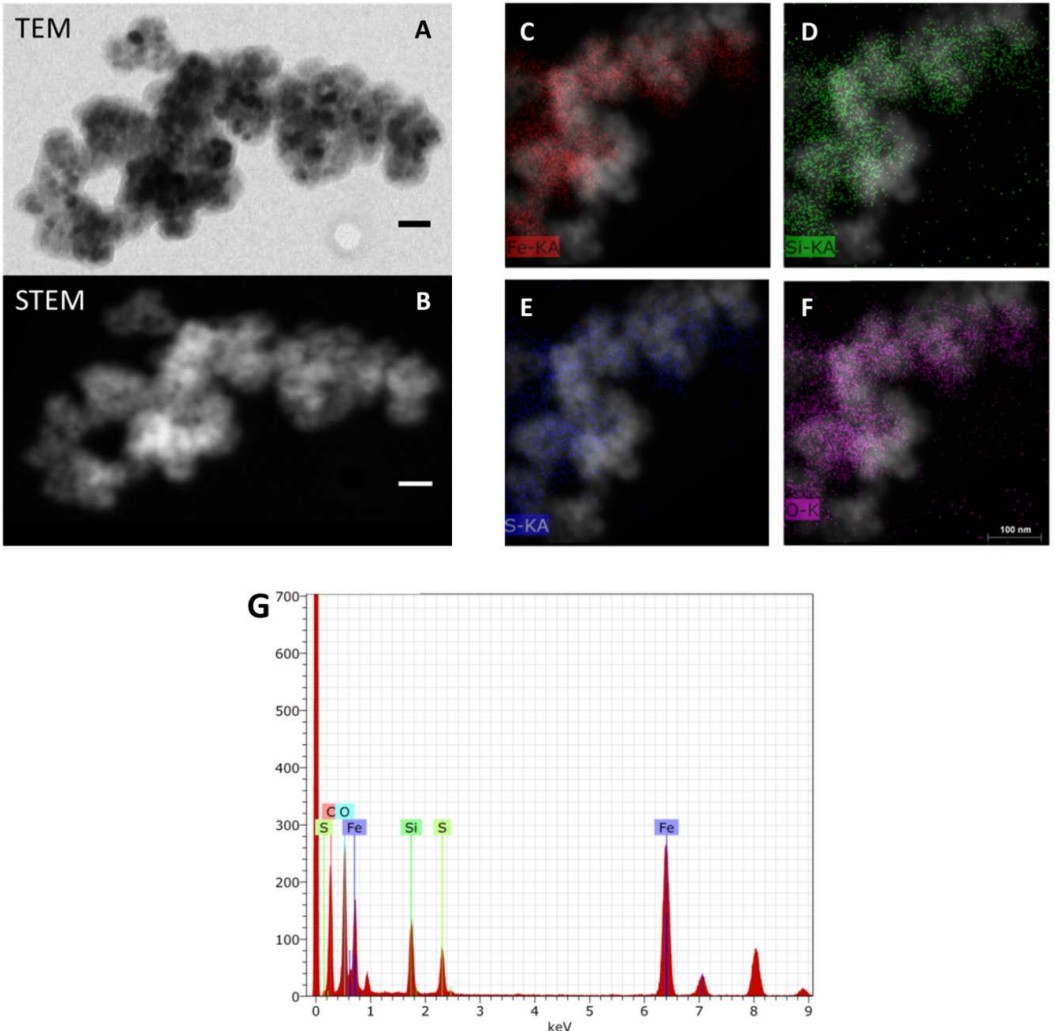

**Figure 3.** (**A**) TEM image of MPTS@MCNP (scale bar 100 nm); (**B**) STEM image of MCNP@MPTS (scale bar 100 nm); (**C–F**) EDX mapping of Fe, Si, S, and O obtained from the respective particles; (**G**) EDX analysis of MPTS@MCNP.

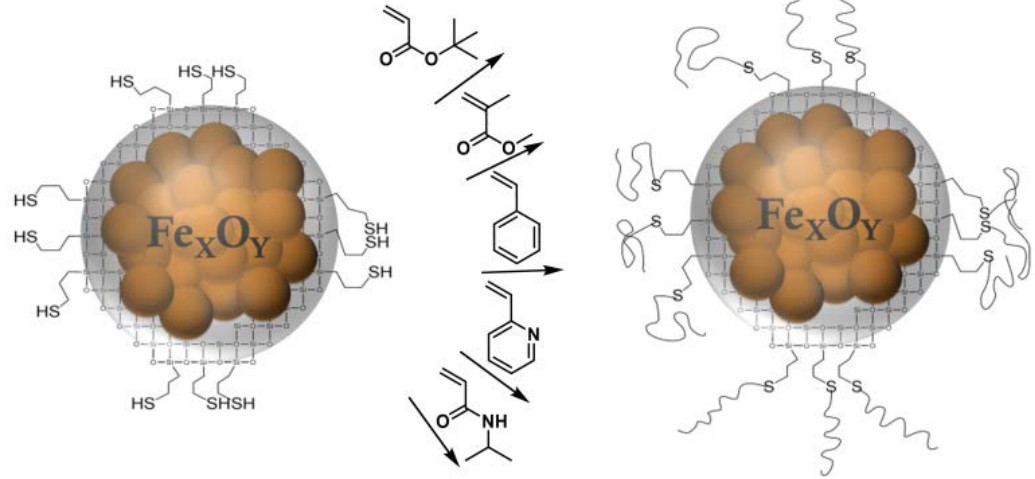

**Scheme 2.** Polymerization of different monomers in the presence of MPTS@MCNP.

By using a photo-initiator (TPO) it was possible to achieve rather short polymerization times of about 20 min. The particles were subsequently magnetically separated from the reaction mixture and the polymers in solution were further analyzed. The SEC traces shown in Figure 4 exhibited a rather broad distribution and the obtained molar masses varied between 9 kg mol$^{-1}$ (P2VP) and 141 kg mol$^{-1}$ (PMMA), depending on the respective monomer. The dispersity was around 2 for all polymers which is typical for a free radical polymerization process.

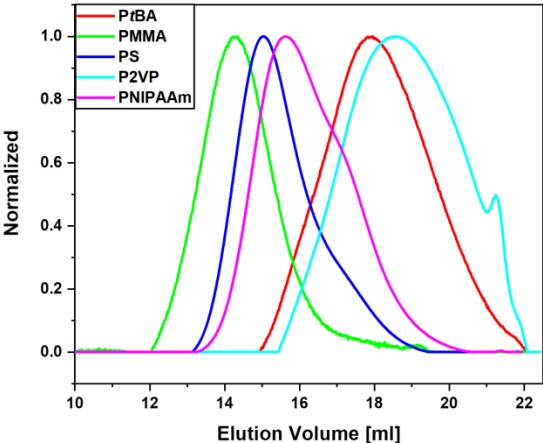

**Figure 4.** SEC elution traces (DMAc/LiCl) of polymers synthesized by free radical polymerization in the presence of MPTS@MCNP as chain transfer agent: red line: P*t*BA ($M_n$ = 8.900 kg mol$^{-1}$; Đ = 2.2); green line: PMMA ($M_n$ = 141.200 kg mol$^{-1}$; Đ = 2.1); blue line: PS ($M_n$ = 58.500 kg mol$^{-1}$; Đ = 1.9); cyan line: P2VP ($M_n$ = 5.600 kg mol$^{-1}$; Đ = 2.3); pink line: PNIPAAM ($M_n$ = 35.400 kg mol$^{-1}$; Đ = 2.0).

**Table 1.** Composition of different reaction mixtures for the free radical polymerization of different monomers in the presence of MPTS@MCNP as chain transfer agent, and SEC-results for the obtained polymers.

| Monomer | Molar Ratios | | Monomer/MPTS@MCNP | $\overline{M_n}$ (kg mol$^{-1}$) | Đ |
|---|---|---|---|---|---|
| | Monomer | TPO | | | |
| *t*BA | 74 | 0.06 | 6.15 g/10 mg | 8.900 (a) | 2.2 (a) |
| MMA | 74 | 0.06 | 4.8 g/10 mg | 141.200 (a) | 2.1 (a) |
| Styrene | 74 | 0.06 | 5 g/10 mg | 58.500 (b) | 1.9 (b) |
| P2VP | 74 | 0.06 | 5.05 g/10 mg | 5.600 (b) | 2.3 (b) |
| NIPAAm | 74 | 0.06 | 5.43 g/10 mg | 35.400 (a) | 2.0 (a) |

(a) Determined by SEC using DMAc/LiCl SEC as eluent and calibrated against PMMA standards; (b) Determined by SEC using DMAc/LiCl SEC as eluent and calibrated against PS standards.

The obtained particles were subsequently washed five times with THF to remove any loosely bound or adsorbed polymer. DLS was used to investigate how the size distribution of MPTS@MCNP changed after the polymerization (Figure 5). The initial radius of 51 nm of MPTS@MCNP in THF increased to values between 174 and 224 nm. We explain this rather drastic size increase by a combination of the polymer shell being formed together with a potential incorporation of several nanoparticles within one polymer shell, e.g., by recombination reactions occurring during radical polymerization. In addition, PMMA@MPTS@MCNP and P2VP@MPTS@MCNP showed a shoulder to higher hydrodynamic radii, which also hint towards some secondary aggregation taking place.

Thermogravimetric analysis (TGA) was performed to quantitatively analyze the amount of bound polymer (Figure 5). The thiol-functionalized particles showed a mass loss of 23.2% attributed to the organic compounds of the siloxane shell. The main weight loss for the particles after application of a polymer coating was obtained in the temperature range of 230–600 °C. Compared to MPTS@MCNP, the onset of the decomposition was shifted to lower temperatures, also indicating the presence of a

polymer shell. One exception was observed for PNIPAAm where the main decomposition occurred between 250–620 °C. In comparison to the thermogravimetric measurement of MPTS@MCNP, we were able to estimate the relative amount of polymer which was between 31% (PMMA) and 21% (P*t*BA) for the majority of polymers. Again, PNIPAAm@MPTS@MCNP presented an exception as here only 11 wt % were found. We used the values derived from TGA measurements to calculate a theoretical shell thickness for each polymer coating. The formula used is given in the Supporting Information and the calculated thicknesses were between 7 nm (PNIPAAm) and 19 nm (PMMA). Please note that the calculated values are based on several assumptions, e.g., the presence of only spherical particles and a mixed core density composed of the MPTS shell and the $Fe_2O_3$ core–hence, these values should be treated as rough estimates. However, TGA confirmed successful immobilization for all polymers at the nanoparticle surface. A control reaction using pristine MCNP and *t*BA as monomer resulted in no observable polymer shell (Figure S2B, Supplementary Materials) indicating that a MPTS shell is mandatory for a successful polymer immobilization.

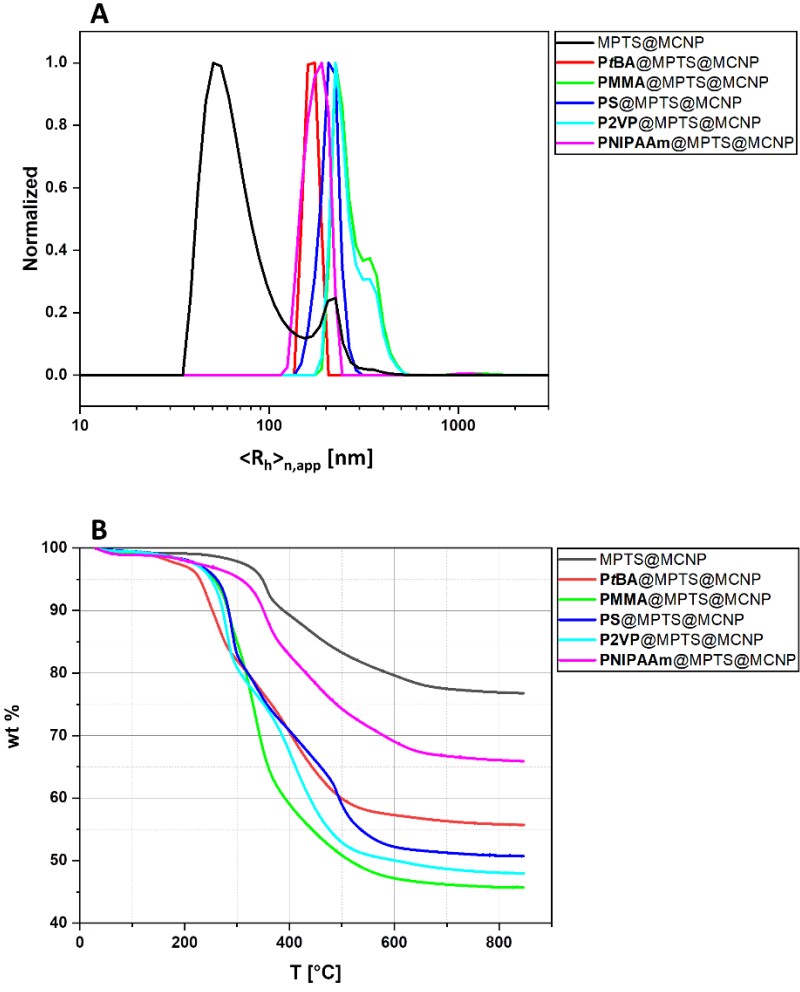

**Figure 5.** (**A**) Number-weighted DLS CONTIN plots of MPTS@MCNP (black line, $\langle R_h \rangle_{n,app}$ = 51 nm), P*t*BA@MPTS@MCNP (red line, $\langle R_h \rangle_{n,app}$ = 174 nm), PMMA@MPTS@MCNP (green line, $\langle R_h \rangle_{n,app}$ = 224 nm), PS@MPTS@MCNP (blue line, $\langle R_h \rangle_{n,app}$ = 206 nm), P2VP@MPTS@MCNP (cyan line, $\langle R_h \rangle_{n,app}$ = 224 nm), PNIPAAm@MPTS@MCNP (pink line, $\langle R_h \rangle_{n,app}$ = 190 nm). (**B**) Thermograms between 50 °C and 850 °C under synthetic air of MPTS@MCNP (black line, 23.2% overall weight loss), P*t*BA@MPTS@MCNP (red line, 44.3% overall weight loss), PMMA@MPTS@MCNP (green line, 54.3% overall weight loss), PS@MPTS@MCNP (blue line, 49.3% overall weight loss), P2VP@MPTS@MCNP (cyan line, 52.0% overall weight loss); PNIPAAm@MPTS@MCNP (pink line, 34.1% overall weight loss).

The TEM micrographs in Figure 6 show the MPTS-coated particles and the core–shell-shell nanoparticles obtained after polymerization. It is evident that the radius of the particles increased after modification, which we explain by partial aggregation and the presence of a thicker shell. However, it is difficult to see any difference in contrast between both shells. Nevertheless, the overall size of the aggregates in the TEM micrographs is within the same range as the values obtained from DLS measurements (between 200 and 400 nm) (Table 2).

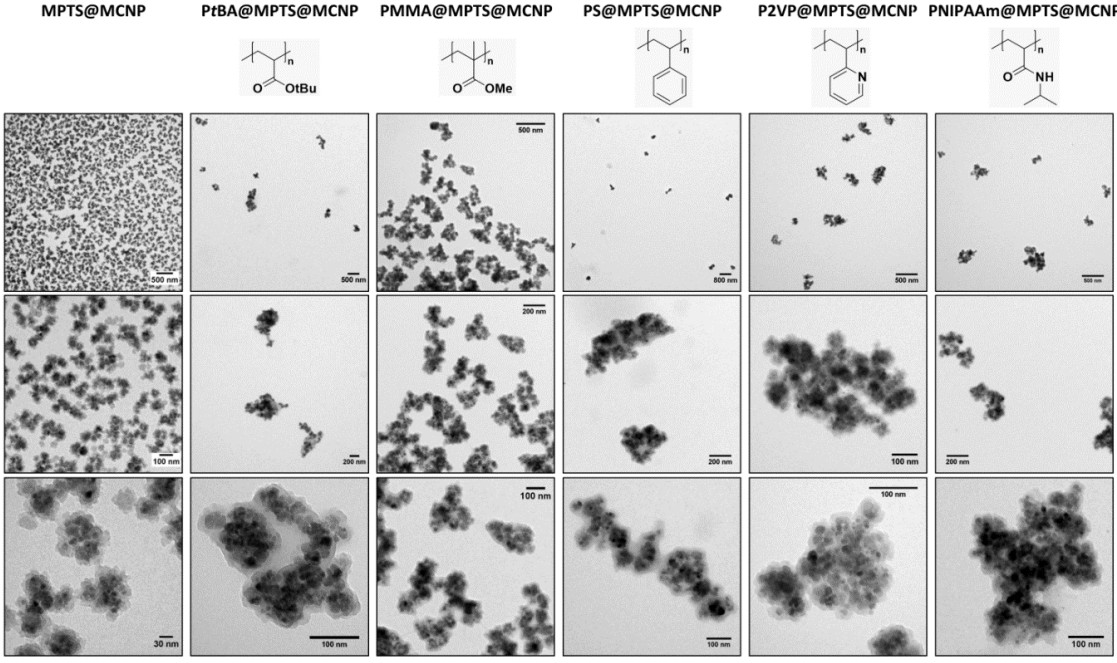

**Figure 6.** TEM micrographs of different polymer@MPTS@MCNP obtained from polymerizations in the presence of MPTS@MCNP as chain transfer agent at different magnification.

**Table 2.** Mass loss between 200 and 800 °C under synthetic air and hydrodynamic radius of Polymer@MPTS@MCNP.

| Sample | wt Loss (%) | $\langle R_h \rangle_{n,app\ max}$ (nm) | Calculated Polymer Shell Thickness (nm) |
|---|---|---|---|
| MPTS@MCNP | 23.2 | 51 | |
| P*t*BA@MPTS@MCNP | 44.3 ($\Delta$ = 21.1) | 174 | 14 (a) |
| PMMA@MPTS@MCNP | 54.3 ($\Delta$ = 31.1) | 224 | 19 (a) |
| PS@MPTS@MCNP | 49.3 ($\Delta$ = 26.1) | 206 | 17 (a) |
| P2VP@MPTS@MCNP | 52.0 ($\Delta$ = 28.8) | 224 | 18 (a) |
| PNIPAAm@MPTS@MCNP | 34.1 ($\Delta$ = 10.9) | 190 | 7 (a) |

(a) Calculated according to Equation S1, Supplementary Materials.

As another method to probe the chemical composition, ATR FT-IR measurements were performed to analyze the functional groups for MPTS@MCNP and polymer@MPTS@MCNP (Figure 7). Whereas MPTS@MCNP did not show any characteristic bands, specific signals for the surface immobilized polymer were observed in all other cases (Table 3). Particles functionalized with carbonyl containing polymers showed characteristic bands around 1720 cm$^{-1}$, while bands for the aromatic ring system (between 1600–1570 cm$^{-1}$ and 1500–1470 cm$^{-1}$) appeared for P2VP and PS. In addition, after immobilization of PNIPAAm an additional band around 1550 cm$^{-1}$ was evident, which can be assigned to the amide functionality. Altogether, both TGA as well as ATR FT-IR confirmed the successful formation of different polymer shells.

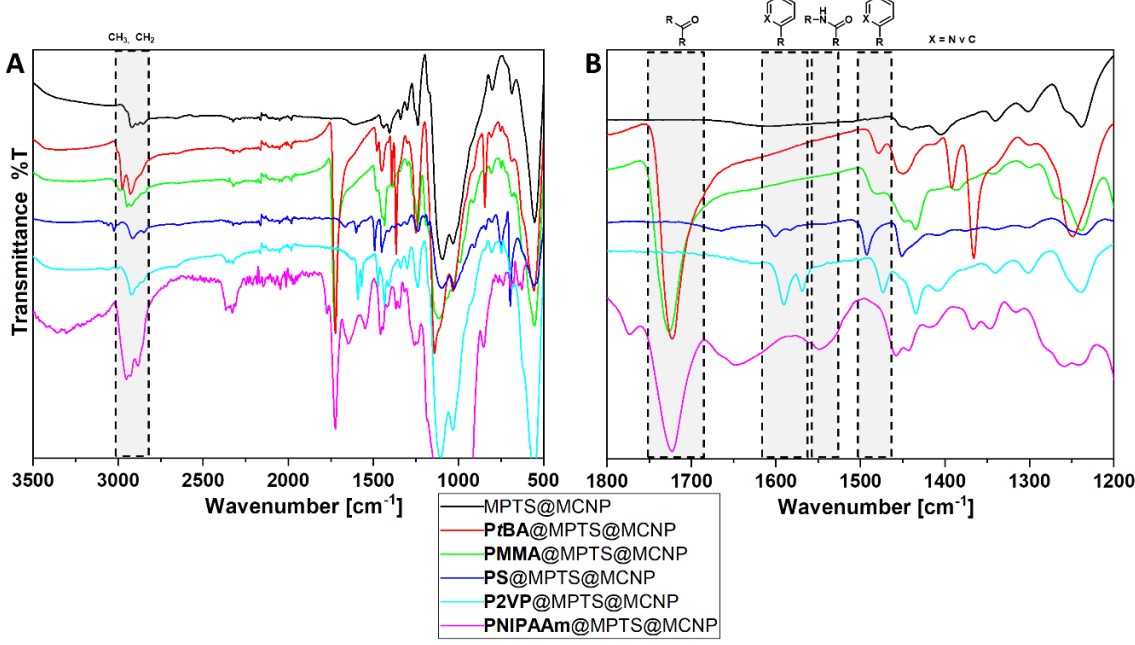

**Figure 7.** (**A**) ATR-FTIR spectra—black line: MPTS@MCNP; red line: P*t*BA@MPTS@MCNP; green line: PMMA@MPTS@MCNP; blue line: PS@MPTS@MCNP; cyan line: P2VP@MPTS@MCNP; pink line: PNIPAAm@MPTS@MCNP. (**B**) Enlargement of the region within the spectra where functional groups can be assigned.

**Table 3.** Assignment of selected IR-bands of polymer@MPTS@MCNP

| Band Assignment | Wavenumbers in cm$^{-1}$ Surface Coating | | | | |
| --- | --- | --- | --- | --- | --- |
| | P*t*BA | PMMA | PS | P2VP | PNIPAAm |
| $\nu_{as}$ (CH$_3$, CH$_2$) | 2975 ms | 2993 m | - | 2926 s | 2953 s |
| $\nu_{as}$ (CH$_3$, CH$_2$) | 2924 ms | 2950 ms | 2912 m | 2336 w | 2882 s |
| $\nu$ (C=O) | 1722 s | 1725 s | - | - | 1721 s |
| $\delta$ (CO–N–H) | - | - | - | - | 1549 m |
| $\nu$ (C–C in–ring) | - | - | 1603 w | 1591 m/1569 s | - |
| $\nu$ (C–C in–ring) | - | - | 1493 ms/1451 ms | 1473 ms | - |

Vibrational mode: $\nu_{as}$—asymmetric stretching vibration; $\nu$—stretching vibration; $\delta$—deformation vibration; Intensities: w—weak; m—medium, ms—medium strong, s—strong.

It could further be observed that the nanoparticles exhibited different solution properties after coating. For example, PNIPAAm/P2VP@MPTS@MCNP were well-dispersed in THF and water, whereas all other particles could mainly be dispersed in THF and other organic solvents. However, an expected temperature response of PNIPAAm@MPTS@MCNP could not be observed. Whereas this was initially surprising, similar findings were made by Wang et al. who synthesized magnetic nanoparticles with a coating of P(PEGMA-co-NIPAAm). Here, also no thermo-responsive behavior was observed, which was explained by aggregation hindrance due to sterical constraints of the polymer chains after surface immobilization [31].

### 3.3. Influence of Particle Concentration on the Polymerization

As the surface bound thiols act as chain transfer agents during the radical polymerization step it can be expected that the nanoparticle concentration has an influence on the polymerization. Therefore, the amount of MPTS@MCNP during a styrene polymerization was varied between 5 and 50 mg and the resulting polymers were investigated by SEC (Figure 8). As expected, the dispersity was decreasing from 9.6 to 2.8 with an increasing amount of MPTS@MCNP, while at the same time the $M_n$

was increasing, both effects of the chain transfer agent concentration. However, taking into account the shape of the SEC elution traces and the fact that the maximum in the elution curve is not constantly decreasing leads us to the conclusion that the influence on the overall dispersity is more pronounced (Table 4).

**Table 4.** SEC results for PS obtained by free radical polymerization in the presence of different amounts of MPTS@MCNP as chain transfer agent

| MPTS@MCNP (mg) | $M_n$ (a) (kg mol$^{-1}$) | Đ (a) |
|:---:|:---:|:---:|
| 5 | 6.900 | 9.6 |
| 10 | 12.800 | 5.2 |
| 20 | 13.300 | 4.5 |
| 50 | 62.600 | 2.8 |

(a) Determined by SEC using DMAc/LiCl SEC as eluent and calibrated against PS standards.

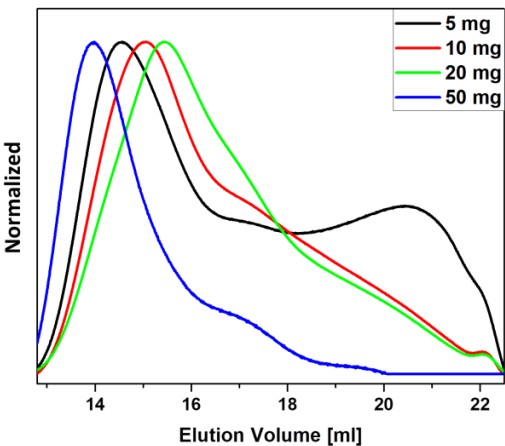

**Figure 8.** SEC elution traces (DMAc/LiCl) of PS synthesized by free radical polymerization in the presence of different amounts of MPTS@MCNP as chain transfer agent: black line: PS obtained in the presence of 5 mg ($M_n$ = 6.900 kg mol$^{-1}$; Đ = 9.6); red line: PS (10 mg, $M_n$ = 12.800 kg mol$^{-1}$; Đ = 5.2); green line: PS (20 mg, $M_n$ = 13.300 kg mol$^{-1}$; Đ = 4.5); blue line: PS (50 mg, $M_n$ = 62.600 kg mol$^{-1}$; Đ = 2.8).

### 3.4. Comparison of Polymer Formed at the Nanoparticle Surface vs. in Solution

We were interested in comparing the characteristics of polymer formed in solution to the material covalently immobilized at the surface of MPTS@MCNP. We therefore exemplarily investigated PS@MPTS@MCNP obtained in the presence of 50 mg MPTS@MCNP where according to TGA data the thickest polymer shell was formed (Figure S3, Supplementary Materials). TEM images show that the particles indeed exhibit a thick shell and form large aggregates (Figure 9A–D), which we so far attribute to the increased concentration of MPTS@MCNP during polymerization. PS@MPTS@MCNP were poorly water soluble and exhibited a foil-like structure upon drying, as well as increased resistance towards 1 M aq. HCl as the formation of an organic shell with increased hydrophobicity reduces the accessibility of the magnetic core. While the pristine particles dissolved within seconds, ultrasonication for several minutes was necessary in case of coated particles.

After PS@MPTS@MCNP were exposed to 1 M HCl/ethyl acetate solution to dissolve the iron oxide core, the siloxane shell was dissolved in a second step using 1 M aq. KOH. The obtained polymer was precipitated and investigated via SEC (Figure 9E). Both elution curves of PS obtained in the polymerization solution and from PS@MPTS@MCNP matched well although at higher elution volumes a shoulder was visible for PS from solution. This is an indication that polymers with lower molar masses might be less represented on the particle surface.

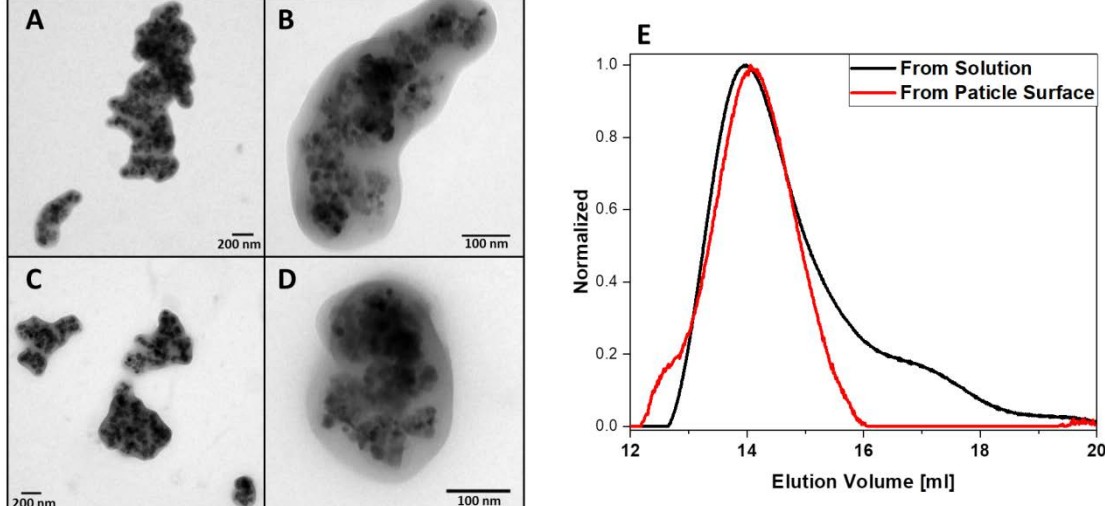

**Figure 9.** (**A–D**) TEM micrographs of PS@MPTS@MCNP; (**E**) SEC elution traces (DMAc/LiCl) of PS synthesized by free radical polymerization in the presence of MPTS@MCNP: black line: PS from reaction solution ($M_\mathrm{n}$ = 62.600 kg mol$^{-1}$; Đ = 2.78); red line: PS obtained from the particle surface ($M_\mathrm{n}$ = 165 kg mol$^{-1}$; Đ = 1.80).

## 4. Conclusions

We demonstrated a simple and straightforward method to covalently immobilize different polymers at the surface of iron oxide MNPs. The obtained hybrid materials were synthesized in a two-step coating process: silanization was first used to immobilize thiol-groups which were used in a second coating step for a grafting through approach in a free radical polymerization. The silanization was optimized in terms of shell thickness and size distribution of the nanoparticles and allowed access to defined nanomaterials which served as a platform for further surface modification. We examined several monomers (MMA, *t*BA, sytrene, 2VP, and NIPAAm) as potential polymer coatings and were able to show that this method allows the formation of a broad variety of polymer shells. The resulting materials exhibited a polymer content of 10–30 wt % and the amount of thiol-functionalized MCNP had a direct effect on the polymerization process. Finally, the mean molecular weight of the surface-immobilized polymer in case of polystyrene corresponded quite well to the material formed in solution during the polymerization. In conclusion, we present a straightforward, time- and cost-efficient method to access a broad variety of covalently anchored polymeric coatings for magnetic nanoparticles.

**Supplementary Materials:** The following are available online at http://www.mdpi.com/2571-9637/3/1/11/s1: Figure S1: TEM micrographs of pristine MCNP, Figure S2: Number-weighted DLS CONTIN plots of MPTS@MCNP in different solvents and thermograms for a control experiment using pristine MCNP for grafting attempts; Figure S3: Thermograms between 50 °C and 850 °C under synthetic air of MPTS@MCNP (black line, 19% overall weight loss), PS@MPTS@MCNP (red line, 50% overall weight loss); Equation S1: Formula used for the calculation of shell thickness according to TGA.

**Author Contributions:** P.B. and F.H.S. conceived this work. P.B. carried out the synthesis, physico chemical characterization, and data analysis. P.B. and F.H.S. jointly discussed results and wrote the manuscript. All authors have read and agreed to the published version of the manuscript.

**Funding:** This research was funded by the Deutsche Forschungsgemeinschaft (DFG, SCHA1640/12-1).

**Acknowledgments:** The authors are grateful to Andreas Weidner (TU Ilmenau) for providing the multicore magnetic nanoparticles. F.H.S. and P.B. gratefully acknowledge financial support through the Deutsche Forschungsgemeinschaft (DFG, SCHA1640/12-1). The TEM facilities of the Jena Center for Soft Matter (JCSM) were established with a grant from the German Research Council (DFG) and the European Fonds for Regional Development (EFRE).

**Conflicts of Interest:** The authors declare no conflict of interest.

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
