# Peer review of "Surface Functionalization of Magnetic Nanoparticles Using a Thiol-Based Grafting-Through Approach"

_surfaces, doi:10.3390/surfaces3010011_

Round 1
Reviewer 1 Report
The manuscript given by Biehl and Schacher reports on the preparation of functional polymers covalently linked to the surface of magnetic iron oxide particle via a chain-transfer agent polymerization strategy. This became possible by immobilization of a thiol linker to the surface of the nano particles. Obtained highly interesting core-shell particles were thoroughly characterized by DLS, TGA, EDX, TEM, ATR-FTIR, while for the freely formed polymers in solution, state-of-the-art polymer analytical tools were used. All these analytical data prove the success of the – from my opinion – versatile and universal approach. The manuscript is well-written and references and recent approaches are carefully summarized. From my point of view this manuscript gives an excellent contribution to MDPI Surfaces taking the following minor inconsistencies and suggestions into account:
- Page 3, line 137: change ‘DMAC’
- Page 4, line 154: The differentiation between the already in literature reported ‘two step procedure’ and the proposed one-step approach is not clear, without reading the given references. Maybe the authors can briefly point out the differences and novelty of their approach more.
- Figure 1: Do the samples reach a plateau and degrade further at 830°C? it seems that there is still a drift and weight loss at the highest temperature.
- Within the manuscript, the sentence ‘reference source not found’ appears, making it hard to sort all the given data. Please change accordingly for the revised version (lines 170, 176, 192, 205, 227, 232, 249).
- Line 169/line 183: Please either name the functional chain transfer shell ‘silica shell’ (line 169) or ‘organic shell’ (line 183). From my point of view it is a hybrid shell material, consisting of a siloxane-species.
- In many Tables, values are given with a comma instead of a dot (for example 8,900 for the molar mass in Table 1). This also counts for the Supporting Information and legend of several figures (e.g. 5,…)
- Table 1: give all units for the data
- Line 249: how was the value of 51 nm obtained? I could not find this value within the DLS measurements (Fig S2???)?
- Figure 5: maybe an improved scaling helps to distinguish between these curves.
- Line 297/298: font style
- Figure 9: Are these scale bars all correct? Also for the zoom-in images?
Author Response
Reviewer #1
The manuscript given by Biehl and Schacher reports on the preparation of functional polymers covalently linked to the surface of magnetic iron oxide particle via a chain-transfer agent polymerization strategy. This became possible by immobilization of a thiol linker to the surface of the nano particles. Obtained highly interesting core-shell particles were thoroughly characterized by DLS, TGA, EDX, TEM, ATR-FTIR, while for the freely formed polymers in solution, state-of-the-art polymer analytical tools were used. All these analytical data prove the success of the – from my opinion – versatile and universal approach. The manuscript is well-written and references and recent approaches are carefully summarized. From my point of view this manuscript gives an excellent contribution to MDPI Surfaces taking the following minor inconsistencies and suggestions into account:
- Page 3, line 137: change ‘DMAC’
We changed DMAC to DMAc.
- Page 4, line 154: The differentiation between the already in literature reported ‘two step procedure’ and the proposed one-step approach is not clear, without reading the given references. Maybe the authors can briefly point out the differences and novelty of their approach more.
Thanks for the hint. We tried to be more precise on this paragraph and added a more detailed description of drawbacks and advantages of the herein presented method (page 4, line 158-163).
- Figure 1: Do the samples reach a plateau and degrade further at 830°C? it seems that there is still a drift and weight loss at the highest temperature.
Despite the slope of the TGA curves in Figure 1, we assume that the decomposition is complete at 830°C. Any additional weight loss should be negligible, as the investigated organic materials should already be completely degraded at that temperature.
- Within the manuscript, the sentence ‘reference source not found’ appears, making it hard to sort all the given data. Please change accordingly for the revised version (lines 170, 176, 192, 205, 227, 232, 249).
We have corrected the cross-references throughout the manuscript. We would like to add that the assistant editor Leona Li advised us that these errors occurred during file conversion.
- Line 169/line 183: Please either name the functional chain transfer shell ‘silica shell’ (line 169) or ‘organic shell’ (line 183). From my point of view it is a hybrid shell material, consisting of a siloxane-species.
Thanks for the suggestion, we have changed all related text elements to siloxane shell.
- In many Tables, values are given with a comma instead of a dot (for example 8,900 for the molar mass in Table 1). This also counts for the Supporting Information and legend of several figures (e.g. 5,…)
Table 1 and Table 4: We changed all commas to dots and with that the units from g mol-1 to kg mol-1.
- Table 1: give all units for the data
Table 1 and Table 4: We added [kg mol-1] as unit for Mn.
- Line 249: how was the value of 51 nm obtained? I could not find this value within the DLS measurements (Fig S2???)?
The value of 51 nm was obtained from Figure 5. (A), which shows the here used MPTS@MCNP in THF. The particles for this measurement were synthesized using a 3:1 ratio of MPTS:MCNP. The measurements shown in Figure S2 were obtained from experiments using MPTS@MCNP obtained from a ratio 4:1. The results were added to the SI to show a general trend for this type of particles. To clarify this, we have included the missing particle information in the SI.
- Figure 5: maybe an improved scaling helps to distinguish between these curves.
Figure 5: We improved the scale to focus on the relevant size range.
- Line 297/298: font style
We have adapted the style to that of the rest of the document.
- Figure 9: Are these scale bars all correct? Also for the zoom-in images?
Figure 9: We rearranged the scale bars and labeled them according to the length they represent to avoid any obscurities. Consequently we deleted the size description in the caption.
Reviewer 2 Report
In general terms, it is an interesting manuscript with some important results and characterizations.
Nevertheless, some minor corrections are needed, mostly about grammar and some missing parts in the explanation of the obtained results:
- Please check english grammar and redaction through the whole article, because it is difficult to understand. Some of them are underlined in yellow.
- Some references are missing, since it does not appear the number, gut the legend "Error! Reference source not found" Please check.
- On page 12, there is a copy of Figure 7 that it is inserted within the text. Please remove it.

Author Response
In general terms, it is an interesting manuscript with some important results and characterizations.
Nevertheless, some minor corrections are needed, mostly about grammar and some missing parts in the explanation of the obtained results:
- Please check english grammar and redaction through the whole article, because it is difficult to understand. Some of them are underlined in yellow.
We have read the article carefully and grammatically revised certain parts.
- Some references are missing, since it does not appear the number, gut the legend "Error! Reference source not found" Please check.
We have revised the article for cross-reference. We would like to add that the assistant editor Leona Li advised us that these errors occurred during file conversion.
- On page 12, there is a copy of Figure 7 that it is inserted within the text. Please remove it.
The figure was removed.